# Sustainable Land-Use Allocation Model at a Watershed Level under Uncertainty

**DOI:** 10.3390/ijerph182413411

**Published:** 2021-12-20

**Authors:** Yao Lu, Min Zhou, Guoliang Ou, Zuo Zhang, Li He, Yuxiang Ma, Chaonan Ma, Jiating Tu, Siqi Li

**Affiliations:** 1College of Marxism, Hubei University, Wuhan 430062, China; 20200053@hubu.edu.cn; 2College of Public Administration, Huazhong University of Science and Technology, Wuhan 430030, China; myx0996@163.com (Y.M.); 17863935943@163.com (C.M.); tujiating1999@163.com (J.T.); withdoca@163.com (S.L.); 3School of Construction and Environmental Engineering, Shenzhen Polytechnic, Shenzhen 518055, China; 4School of Public Administration, Central China Normal University, Wuhan 430079, China; zhangzuocug@163.com; 5School of Urban Construction, Yangtze University, Jingzhou 434023, China; heli@yangtzeu.edu.cn

**Keywords:** sustainable land use, ecological environment protection, interval fuzzy two-stage stochastic model, South Four Lake watershed, eco-environmental constraints

## Abstract

Land-use allocation models can effectively support sustainable land use. A large number of studies solve the problems of land-use planning by constructing models, such as mathematical models and spatial analysis models. However, these models fail to fully and comprehensively consider three uncertain factors of land-use systems: randomness, interval and fuzziness. 33Therefore, through the study of the watershed land-use system, this paper develops a land-use allocation model considering the regional land–society–economy–environment system under uncertain conditions. On the basis of this model, an interval fuzzy two-stage random land-use allocation model (IFTSP-LUAM) combining social, economic and ecological factors is proposed to provide sustainable development strategies at the basin level. In addition, the proposed IFTSP-LUAM takes into account the above three uncertainties and multistage, multiobjective, dynamic, systematic and complex characteristics of typical land-use planning systems. The results showed that the model considers more socioeconomic and ecological factors and can effectively reflect the quantitative relationship between the increase in economic benefits and the decrease in environmental costs of a land-use system. The model was applied to land-use planning of Nansihu River Basin in Shandong Province. The results provided a series of suitable land-use patterns and environmental emission scenarios under uncertain conditions, which can help the watershed environmental protection bureau and watershed land-use decision-makers to formulate appropriate land-use policies, so as to balance social and economic development and ecological protection. The simulation results can provide support for an in-depth analysis of land-use patterns and the trade-off between economic development and ecological environment protection.

## 1. Introduction

Land is one of the most valuable natural resources in the world. Due to rapid population growth, urbanization and industrialization, China is suffering from ecological imbalance and environmental pollution, which have brought a series of social problems to its people [1,2]. Sustainable development is a good strategy to solve this problem. Many scholars are focusing on exploring methods to promote the sustainable use of land resources both in China and all over the world [3]. Many theories, technologies and models have been proposed to solve the challenge of sustainable land use [4].

For many years, many scholars have proposed effective models for land-use planning. Models about land-use planning (LUP) fall into four categories, namely the mathematical programming models, the spatial optimization models, the intelligent models and the coupling models. Research on mathematical programming models for LUP started earlier, and they are also the most widely used models at present [5,6,7,8,9,10,11,12]. This kind of model mainly regards the LUP problem as an objective optimization problem, and it generally solves the optimal solution by designing an objective function and a constraint condition, so as to achieve the optimization objective. At present, the most widely used mathematical programming model is the general linear programming model. For example, Mendoza [5] proposed a linear programming land-use allocation model. Aerts et al. [6] used a linear integer programming method to complete multisite land-use allocation. Santé and Crecente and Gong et al. [7,8] developed a multiobjective linear programming land-use allocation model. This kind of model has the characteristics of clear objectives, wide application and easy calculation. However, this kind of model requires high accuracy of parameters, lacks flexibility and has difficulty in effectively dealing with nonlinear problems in land-use systems. In view of the shortcomings of general linear programming models, scholars have gradually introduced nonlinear programming models and uncertain mathematical models such as fuzzy programming [9,10,11], random programming [12,13] and interval programming [14,15]. For instance, Haque and Asami [16] reported a nonlinear programming model for urban land-use allocation. Wang et al. [17] described an interval multiobjective linear programming model for land-use allocation practice. Dai and Li [18] studied an interval multistage stochastic programming model for agricultural crop land-use allocation. Elalamy et al. [19] developed a bio-economic programming model for land-use allocation. Spatial optimization models mainly allocate various types of land use in space [20,21,22,23,24,25,26,27,28,29,30,31,32,33]. Their goal is to allocate land-use spatial units to land types with high suitability and matching degrees. This process is extremely complicated and requires coordination of economic, ecological and spatial objectives. At present, the main types are geographic information system (GIS)-based spatial allocation models [34,35,36], cellular automata (CA) models, the Conversion of Land Use and Its Effects (CLUE) model, etc. For example, Cromley and Hanink [20,21] proposed a raster-based GIS method for solving multicriteria, multiobjective land-use allocation problems. Santé-Riveira et al. [23] described a GIS-based spatial planning decision system for rural land-use allocation. Verburg et al. [31] used the CLUE model to forecast the near-future land-use changes. Ke et al. and Zheng et al. [32,33] proposed a CA-based land-use allocation model (which was called the LANDSCAPE model). As intelligent models, aiming at the complex problems in land-use systems, such as multiobjective, nonlinear and stochastic problems, scholars have introduced an allocation model based on intelligent algorithms [37,38,39,40,41,42,43,44,45,46]. For instance, Aerts et al. [37] presented a goal-programming model involving simulated annealing and genetic algorithms and solved multisite land-use allocation problems using the model. Sharawi [38] reported a Little–Mirrlees–Squire–van der Tak (LMST) method for land-use allocation. Ahmadi et al. [39] and Fotakis and Sidiropoulos [40] used a genetic algorithm to achieve land-use allocation, while Santé-Riveira et al. [41] proposed a simulated annealing algorithm for land-use allocation. Eldrandaly [42] presented a gene expression programming (GEP) model for multisite land-use allocation. Huang et al. [43] developed an improved artificial immune system for multiobjective land-use allocation. Liu et al. [44] presented a multitype ant colony optimization model for land-use allocation. Cao and Ye [45] presented a coarse-grained parallel genetic algorithm for land-use allocation. Zhou et al. [46] used a dimidiate pixel model to conduct land-use allocation. Coupling models for the LUP address the issues of land-use quantity structure and spatial allocation [47,48,49,50,51,52,53,54,55,56,57,58]. Many researchers have coupled the land-use quantity allocation model with the spatial allocation model, so as to make the land-use quantity be implemented in space and further improve the comprehensive benefits of the land-use system. Chuvieco and Zhang et al. [49,50] presented a hybrid linear programming coupling GIS model for land-use allocation. Ma and Zhou [51] developed a GIS-incorporated interval fuzzy linear programming model for land-use allocation. Huang and Song [52] proposed a land-use spatial optimum allocation model that involved the coupling of a multiagent system with the shuffled frog leaping algorithm. Liu et al. [54] coupled the particle swarm optimization model and multiobjective programming method for rural spatial land-use allocation. Liu et al. [44] integrated the system dynamics method and particle swarm optimization model for solving the land-use allocation problems.

The above models can effectively deal with the problems in LUP, but there are still some deficiencies: (1) The ability of the above models to systematically consider and analyze the uncertain factors in land-use systems needs to be improved. There are many uncertain factors in land-use systems, which are generally divided into random factors, fuzzy factors and interval factors [45,46,47,48]. At present, some studies have considered the above factors when studying the LUP problem. For example, Zhou et al. [59] proposed an interval fuzzy land-use allocation model (IFLAM) for Beijing (China) that contained certain environmental and ecological considerations. Li et al. [60] proposed a copula-based interval stochastic programming model for land-use allocation. However, the above studies obviously failed to fully consider the three types of uncertain factors and integrated the three types of uncertain factors when building the model. (2) Ecological environmental factors have been included in many of the previously reported land-use planning models [61,62]. However, some important factors, such as pollution emissions (for example, air pollution and solid waste) and ecologically limiting activities (such as soil erosion and fertilizer application) were neglected in these models. 

In short, traditional land-use planning models can solve land-use planning problems considering various factors (economical, social, environmental, ecological, legal and policy factors), multiple scales (urban, city, region, watershed and country) and three uncertainties (discrete intervals, possibilities and fuzzy sets). However, a complete uncertain land-use planning model, which couples the three uncertain mathematical models and considers many more ecological environmental factors, still needs to be developed. With this backdrop, the current study proposes an interval fuzzy two-stage stochastic land-use planning model (IFTSP-LUPM) for land-use allocation and ecological environmental analysis/management in South Four Lake watershed, Shandong, China. The modeling results can support the quantitative analysis of land-use patterns and the trade-off between economic development and ecological environmental protection.

## 2. Study Area

### 2.1. General Situation

South Four Lake watershed is located in the southwest of Shandong province, China (116°34′–117°21′ E, 34°27′–35°20′ N) (Figure 1). The watershed includes four cities ( Jining, Heze, Zaozhuang and Ningyang) and four lakes ( Weishan, Zhaoyang, Dushan and Nanyang). The watershed area is around 30,230 km^2^ with a maximum water storage of 53.6 × 10^9^ m^3^. The maximum lake area is 1266 km^2^, which occupies around 45% of the fresh-water area of Shandong, China. South Four Lake is the biggest fresh-water resource in Shandong and the North China Plain. The lake is also very important for the Eastern route of South-to-North Water Transfer Project of China as it is the most important water channel and the main impounded lake.

### 2.2. Socioeconomic Situation

The population of the watershed has grown from 19.46 × 10^6^ to 23.61 × 10^6^ in the past 16 years (2000–2016) with an average annual growth rate of 0.412%, which is approximately equal to the mean value for China as a whole. The gross domestic product (GDP) of the watershed grew from 3903.94 × 10^9^ Renminbi (RMB) to 10684.53 × 10^9^ RMB in the past 10 years, which is much lower than the other regions of Shandong, China. The urbanization rates of the four cities in this watershed are 32.15%, 34.06%, 22.02% and 20.89%, which are also lower than the average level of Shandong, China. In the early reform period of China, the internal proportion of the three biggest industries (agriculture, industry and tertiary industry) was 57:27:16, whereas in 2000, the proportion changed to 25:41:34. Moreover, in 2016, the proportion of the three big industries changed to 17:50:33. It can be seen that the proportion of agriculture is reducing throughout this time period, whereas the proportion of industry is increasing. This change explains the GDP growth and implies the increasingly serious ecological environmental problems.

### 2.3. Land-Use Status

From 2005 to 2015, significant changes have taken place in the land structure of South Four Lake watershed. Firstly, cultivated land has decreased from 18365.11 to 17404.35 km^2^ with an average annual decrease of 87.34 km^2^. Secondly, the garden land has gone through small increases on an annual basis and changed from 634.69 to 739.50 km^2^. The increase in garden land has come from the cultivated land. The main reason is that the economic benefits of planting vegetables are much higher than those of planting grains in the watershed. Thirdly, the forest land has increased year by year with an increment of 291.14 km^2^ per year. This is due to the policy of returning farmland to forest in the watershed. However, it should be noticed that the forest coverage in the watershed was 7.0% in 2015, which is much lower than the average value in China (21.63%). Fourthly, grassland in the watershed is scarce; therefore, the change in grassland can be ignored. Fifthly, the construction land (especially, the industrial and mining lands) has increased from 3415.07 to 4038.11 km^2^ with a variation rate of 18.24%, which has led to serious environmental pollution and ecological damage.

### 2.4. Ecological Environmental Status

South Four Lake watershed is one of the most polluted areas in China, especially with regard to water pollution. Industrial structural pollution and city comprehensive pollution are very serious, leading to a sharp contrast between economic development and environmental protection. Extensive economic activity is the main growth pattern in the watershed. There are many industries causing environmental pollution in the watershed, including the paper industry, chemical industry, brewing industry and printing and dyeing industry. Fish and shrimp have disappeared in the 53 rivers of the watershed. Moreover, chemical oxygen demand (COD) in some areas of the South Four Lake has exceeded the value of 1000 mg/L. Furthermore, some inappropriate human activities (such as land reclamation from the lake for agriculture, overfishing and waste discharge into the lake) have led to serious damage to the ecological system of the watershed. For example, approximately 300 km^2^ of wetland has disappeared in the watershed, and the bearing capacity of the natural ecosystem and the restoration of water bodies and their self-purification capacities have been seriously weakened.

The watershed is characterized by fast economic development, unreasonable land-use structure and serious ecological environmental problems.

## 3. Interval Fuzzy Two-Stage Stochastic Land-Use Allocation Model for South Four Lake Watershed

Land-use allocation helps achieve the optimized economical, social, environmental and ecological objectives in the planning area using land and sustainable development theory, which depends on proper scientific methods and management techniques [63]. A typical land-use allocation system has the following four main characteristics: (i) Dynamic: Physical geographical factors, human socioeconomic factors, technological factors, ecological environments and government policies of the land-use system will change over time. Therefore, the structure and function of land will also change over time. This leads to remarkable dynamic characters of the land-use allocation system. (ii) Systematic and comprehensive: Every land type cannot exist separately as different land-use types are mutually interdependent. This interaction among various land-use practices exists due to various geographical factors (climate, landform, hydrology and biology), socioeconomic human factors (policies, population, GDP, labor, transportation, industry and travel) and ecological environmental factors (air pollution, wastewater, solid waste, soil erosion, grass coverage and application of fertilizer). These factors depend on the systematic and comprehensive nature of a land-use allocation system. (iii) Uncertainty: There are various uncertainties in the land-use allocation system, such as intervals, probabilities and fuzzy sets [64]. For example, the land price will float within an interval range. The variation of environmental capacity may accord with a normal distribution. The social and economic conditions could be fuzzy information. When establishing a land-use allocation, these uncertainties should be considered. (iv) Multiobjective: In order to plan a land-use system, there may be more than one objective. Social, economic, environmental and ecological objectives may be considered simultaneously. These characteristics constitute the complexity of a land-use allocation system (Figure 2). Based on the interaction of the system components shown in Figure 2, a conceptual model can be established as follows:

The objective function is maximum economic benefit from the land-use system. It subject to economic, social, environmental and land-use suitability constraints.

The independent variables are the land areas of various land-uses. Furthermore, the proposed conceptual model is detailed based on the coupling of some existing land-use allocation models. Due to this reason, some new constraints are proposed as shown in Table 1.

Based on IFTSP model and mathematical modeling experiences, the proposed conceptual model can be transformed into an IFTSP-LUAM.

### 3.1. Economic Objective

The land-use economic objective should consider microeconomic cost–benefit analysis [71]. In the proposed model, the economic objective is the net benefit produced from the land-use allocation system, which consists of revenues from industries attached to the corresponding land-use types minus the government investment that consists of the waste-tackling cost of commercial/industrial/agricultural/transportation/residential lands, the maintenance cost of water land/unused land and the developing cost of unused land. For example, the commercial industry is attached to commercial land, and therefore, the revenues from commercial land come from the commercial industry (the revenues from the industries that do not occupy land (e.g., financial industry) are not considered). Moreover, the agricultural industry is attached to agricultural land, such as cultivated land, forest land and grassland. Therefore, the revenues from agricultural land come from the agricultural industry. Similarly, waste-tackling cost from industrial land comes from the waste discharge of industries attached to the industrial land. Therefore, the economic objective function can be expressed as Equation (1). The symbols of Equation (1) are described in Table 2.
(1)$$\begin{array}{cc} Max\;{\mathrm{NBL}}^{\pm } & ≅\sum_{k=1}^{3}\sum_{j=1}^{5}\sum_{t=1}^{3}\left(UB_{j,\;k,t}^{\pm }\times x_{j,\;k,t}^{\pm }\right)+\sum_{t=1}^{3}\left(UB_{j=6,\;k,t}^{\pm }\times x_{j=6,\;k,t}^{\pm }\right)\\ & -\sum_{k=1}^{3}\sum_{j=1}^{5}\sum_{t=1}^{3}\left[\left({\mathrm{UWTC}}_{j,\;k,t}^{\pm }+{\mathrm{USTC}}_{j,\;k,t}^{\pm }+{\mathrm{UGTC}}_{j,\;k,t}^{\pm }+{\mathrm{UWSC}}_{j,\;k,t}^{\pm }+{\mathrm{UESC}}_{j,\;k,t}^{\pm }\right)\times x_{\;j,\;k,t}^{\pm }\right]\\ & -\sum_{j=6}^{7}\sum_{t=1}^{3}\left({\mathrm{UMC}}_{j\;,\;t}^{\pm }\times x_{j,\;t}^{\pm }\right)-\sum_{t=1}^{3}\left({\mathrm{UDC}}_{j\;=\;8,\;t}^{\pm }\;\times \;x_{j\;=\;8,\;t}^{\pm }\right) \end{array}$$

### 3.2. Economic Constraints

(i)Government Investment Constraint

In South Four Lake watershed, all costs will be afforded by the government investment. Therefore, the government investment constraints can be expressed using Equation (2).
(2)$$\begin{array}{l} \sum_{k=1}^{3}\sum_{j=1}^{5}\sum_{t=1}^{3}\left[\left({\mathrm{UWTC}}_{j,\;k,t}^{\pm }+{\mathrm{USTC}}_{j,\;k,t}^{\pm }+{\mathrm{UGTC}}_{j,\;k,t}^{\pm }+{\mathrm{UWSC}}_{j,\;k,t}^{\pm }+{\mathrm{UESC}}_{j,\;k,t}^{\pm }\right)\times x_{j,\;k,t}^{\pm }\right]\\ +\sum_{j=6}^{7}\sum_{t=1}^{3}\left({\mathrm{UMC}}_{j\;,\;t}^{\pm }\times x_{j,\;t}^{\pm }\right)+\sum_{t=1}^{3}\left({\mathrm{UDC}}_{j\;=\;8,\;t}^{\pm }\;\times \;x_{j\;=\;8,\;t}^{\pm }\right)\underset{\approx }{\leq }{\mathrm{MGI}}^{\pm } \end{array}$$

(ii)Grain Input–Output Constraint

Grain security is the main issue of South Four Lake watershed. In the proposed model, grain is produced by the agricultural land. The grain production should meet the demand of the South Four Lake watershed, which is given by Equation (3).
(3)$$\sum_{k\;=\;1}^{3}\sum_{t=1}^{3}\left({\mathrm{UGP}}_{j\;=\;4,\;k,t}^{\pm }\;\times \;x_{j\;=\;4,\;k,t}^{\pm }\right)\underset{\approx }{\geq }{\mathrm{DGP}}^{\pm }$$

(iii)Water Production Input–Output Constraint

In South Four Lake watershed, water production should meet the demand in the watershed and satisfy the demand of adjacent big cities, such as Jinan and Qingdao cities. In the proposed model, the water production is provided by the water land, as given by Equation (4).
(4)$$\sum_{t=1}^{3}\left({\mathrm{UWP}}_{j\;=\;6,\;t}^{\pm }\;\times \;x_{j\;=\;6,\;t}^{\pm }\right)\underset{\approx }{\geq }{\mathrm{DWP}}^{\pm }$$

(iv)Available Water Consumption Constraint

All land-uses need water. Water supply comes from the rivers and lakes in the South Four Lake watershed. In the proposed model, water consumption of all land-uses should not exceed the available water supply. This condition is represented by Equation (5).
(5)$$\sum_{k\;=\;1}^{3}\sum_{j=1}^{5}\sum_{t=1}^{3}\left({\mathrm{UWC}}_{j\;,\;k,\;t}^{\pm }\;\times \;x_{j,\;k,\;t}^{\pm }\right)+\sum_{j=7}^{8}\sum_{t=1}^{3}\left({\mathrm{UWC}}_{j,\;t}^{\pm }\;\times \;x_{j,\;t}^{\pm }\right)\underset{\approx }{\leq }{\mathrm{AWS}}^{\pm }$$

(v)Available Electricity Power Consumption Constraint

Similarly, all land-uses need electricity power. The total electricity power consumption should not exceed the available supply capacity, which can be represented using Equation (6).
(6)$$\sum_{k\;=\;1}^{3}\sum_{j=1}^{5}\sum_{t=1}^{3}\left({\mathrm{UEC}}_{j,\;k,\;t}^{\pm }\;\times \;x_{j,\;k,\;t}^{\pm }\right)+\sum_{j=7}^{8}\sum_{t=1}^{3}\left({\mathrm{UEC}}_{j,\;t}^{\pm }\;\times \;x_{j,\;t}^{\pm }\right)\underset{\approx }{\leq }{\mathrm{AES}}^{\pm }$$

### 3.3. Social Constraints

(i)Land Carrying Capacity Constraint

In South Four Lake watershed, the land carrying capacity (LCC) is limited. The maximum population in a unit area should not exceed the maximum LCC in a unit area. This constraint is represented by Equation (7).
(7)$$\mathrm{PP}/\left(\sum_{k\;=\;1}^{3}\sum_{j=1}^{5}\sum_{t=1}^{3}\;x_{j,\;k,\;t}^{\pm }+\sum_{j=6}^{8}\sum_{t=1}^{3}\;x_{j,\;t}^{\pm }\right)\underset{\approx }{\leq }{\mathrm{MLCC}}^{\pm }$$

(ii)Available Labor Constraint

In South Four Lake watershed, all industries attached to land uses need labor. The planning labor in South Four Lake watershed should not exceed the available labor, which is given by Equation (8).
(8)$$\sum_{k\;=\;1}^{3}\sum_{j=1}^{5}\sum_{t=1}^{3}\left(PLU_{j\;,\;k,\;t}^{\pm }\;\times \;x_{j,\;k,\;t}^{\pm }\right)+\sum_{j=6}^{8}\sum_{t=1}^{3}\left({\mathrm{PLU}}_{j,\;t}^{\pm }\;\times \;x_{j,\;t}^{\pm }\right)\underset{\approx }{\leq }{\mathrm{AL}}^{\pm }$$

### 3.4. Land Suitability Constraint

Land suitability assessment is indispensable for land-use allocation. Land areas of some types of land-uses (*j* = 1–5) should accord with the results of land suitability assessment (water land, landfill and unused land do not need land suitability assessment). This constraint is represented by Equation (9).
(9)$$\sum_{j\;=\;1}^{5}\sum_{t=1}^{3}x_{j,\;t}^{\pm }\underset{\approx }{\leq }\sum_{j\;=\;1}^{5}\sum_{t=1}^{3}{\mathrm{HSL}}_{j,\;t}^{\pm }$$

### 3.5. Environmental Constraints

(i)Wastewater Treatment Capacity Constraint

In the model, wastewater produced by some land types (*j* = 1–5) should not exceed the wastewater treatment capacity in the South Four Lake watershed. This constraint is represented by Equation (10).
(10)$$\sum_{k\;=\;1}^{3}\sum_{j=1}^{5}\sum_{t=1}^{3}\left({\mathrm{WDF}}_{j\;,\;k,\;t}^{\pm }\;\times \;x_{j,\;k,\;t}^{\pm }\right)\leq {\mathrm{WPC}}^{p}$$

(ii)Solid Waste Treatment Capacity Constraint

Similarly, solid waste produced by some land types (*j* = 1–5) should not exceed the solid waste treatment capacity and solid waste handling capabilities of the landfill in South Four Lake watershed. This constraint is given by Equation (11).
(11)$$\sum_{k\;=\;1}^{3}\sum_{j=1}^{5}\sum_{t=1}^{3}\left({\mathrm{SDF}}_{j\;,\;k,\;t}^{\pm }\;\times \;x_{j,\;k,\;t}^{\pm }\right)-\sum_{t=1}^{3}\left({\mathrm{SHL}}_{j\;=7,\;t}^{\pm }\;\times \;x_{j=7,\;t}^{\pm }\right)\leq {\mathrm{STC}}^{p}$$

(iii)Air Pollutant Discharge Capacity Constraint

Air pollutants produced by some land types (*j* = 1–5) should not exceed the air pollutant discharge capacity in South Four Lake watershed. This constraint is given by Equation (12).
(12)$$\sum_{k\;=\;1}^{3}\sum_{j=1}^{5}\sum_{t=1}^{3}\left({\mathrm{ADF}}_{j\;,\;k,\;t}^{\pm }\;\times \;x_{j,\;k,\;t}^{\pm }\right)\leq {\mathrm{ADC}}^{p}$$

### 3.6. Ecological Constraints

(i)Available Soil Erosion Constraint

Soil erosion (SE) must be considered in the watershed agricultural land-use planning. In the proposed model, the speed and impacts of agricultural land SE should be considered. The planning agricultural land SE area should not exceed the available SE area in South Four Lake watershed. This constraint is given by Equation (13).
(13)$$\sum_{k\;=\;1}^{3}\sum_{t=1}^{3}\left({\mathrm{SER}}_{j=3\;,\;k,\;t}^{\pm }\;\times \;x_{j=3,\;k,\;t}^{\pm }\right)\leq {\mathrm{ASE}}^{p}$$

(ii)Fertilizer Consumption Constraints

The agricultural land involves a key problem, which is the application of fertilizer. Fertilizer supply is limited in South Four Lake watershed. In the proposed model, the consumption of the fertilizer should not exceed the maximum fertilizer consumption in South Four Lake watershed. This constraint is given by Equation (14).
(14)$$\sum_{k\;=\;1}^{3}\sum_{t=1}^{3}\left({\mathrm{FCU}}_{j=3\;,\;k,\;t}^{\pm }\;\times \;x_{j=3,\;k,\;t}^{\pm }\right)\leq {\mathrm{MFC}}^{p}$$

### 3.7. Technical Constraints

(i)Total Land Area Constraint


(15)
$$\sum_{k\;=\;1}^{3}\sum_{j=1}^{5}\sum_{t=1}^{3}\;x_{j,\;k,\;t}^{\pm }+\sum_{j=6}^{8}\sum_{t=1}^{3}\;x_{j,\;t}^{\pm }={\mathrm{TLA}}^{\pm }$$


(ii)Non-negative Constraint


(16)
$$x^{\pm }\geq 0$$


### 3.8. Parameters of IFTSP-LUAM

Nomenclature for parameters and variables can be found in Table A1 in Appendix A. Parameters of IFTSP-LUAM fall into four types, namely the benefit/cost parameters, the land-use suitability assessment parameters, the social/economical parameters and the ecological/environmental parameters. Benefit/cost parameters can be obtained from the land price assessment and land valuation, where the basic data can be obtained from the Overall Plan of Land Utilization of South Four Lake watershed (2020–2035) (Table A2). Furthermore, the land-use suitability assessment parameters can be obtained from GIS technology (Table A3). The social/economical parameters can be forecasted using the data from the Statistical Yearbook of South Four Lake watershed (1995–2015) (Table A4). Moreover, the ecological/environmental parameters can be forecasted using the data from the Environmental Conditions Bulletin of South Four Lake watershed (1995–2015) (Table A4).

### 3.9. Solving the Model

Using the interactive algorithm [46,49], the IFTSP-LUAM can be transformed into two deterministic submodels, which correspond to the upper and lower bounds for the desired objective function value under different *p* levels. (Table A5) Based upon the arithmetic programming performed in MATLAB, the model can be solved to obtain a series of land-use patterns, environmental emission scenarios and ecological results.

## 4. Results and Discussion

### 4.1. Optimized Land-Use Patterns

Figure 3 and Figure 4 show the interval results from IFTSP-LUAM under different *p* levels and *t* periods under high land suitability level (*k* = 1). Optimized land-use patterns have been obtained under different scenarios (Table 3). 

Under high land suitability level (*k* = 1), period of 2016–2020 (*t* = 1) and *p* of 0.01, the system benefit had the value of [6.58, 7.32] × 10^12^ RMB. It can be seen that the model generated a series of interval results for various land-use patterns and system benefits. The interval results provided two extreme values for each dataset (the lower bound value and the upper bound value of the variables and the objective). Interval values do not provide the distribution of variables and objective, though they can reflect the amplitude of variations. Therefore, the interval results can describe the variation characteristics and can effectively support the scenario analysis because various planning schemes can be generated by selecting a random value between the lower bound and upper bound according to the demand. In order to generate a detailed and common land-use planning scheme for the variation trend analysis, the average value of the intervals was selected. 

### 4.2. Relationship between the Land Suitability Level and the System Benefit

Figure 5 shows the relationship between land suitability level and system benefit. It can be seen that, when *k* was unity, the system benefit was [6.28, 7.32] × 10^12^ RMB, whereas when *k* = 2, the system benefit changed to [5.39, 6.02] × 10^12^ RMB. For *k* = 3, the system benefit changed to [3.34, 4.01] × 10^12^ RMB. The results indicate that the land suitability had a noteworthy influence on system benefit. The land suitability level in South Four Lake watershed was influenced by many factors, including elevation, slope, aspect, soil organic matter content, soil acidity and alkalinity, thickness of soil, soil texture, distance to water, distance to main road and distance to CBD. These factors had an impact on all types of land-uses. Therefore, the benefit from the land-use system was influenced by these factors with respect to the land suitability levels. The quantitative results from the IFTSP-LUAM could help land managers provide an exact insight into the relationship between the land suitability level and the benefit from the land-use system.

### 4.3. Trade-Off between the Economic Development and the Ecological Environmental Protection

Figure 6 shows that the system benefit and the *p* levels have a relationship of positive correlation. When *p* = 0.01, the system benefit was [6.58, 7.32] × 10^12^ RMB, whereas when *p* = 0.05, the system benefit changed to [7.54, 7.69] × 10^12^ RMB. Moreover, when *p* = 0.10, the system benefit increased to [8.26, 9.01] × 10^12^ RMB and for *p* = 0.15, the system benefit took the value of [9.98, 10.68] × 10^12^ RMB. Furthermore, the *p* level represents the probability of violating the ecological environmental constraints. Any change in *p* level would yield different waste management capacities and, therefore, result in different land-use patterns and different system benefits. The results showed that, if the environmental risk was increased to 10%, the system benefit increased to 2.0×10^12^ RMB.

Since the *p* levels represent the probabilities at which the environmental constraints will be violated, the relationship between the system benefit and *p* demonstrates a trade-off between the economic efficiency and the system risk. An increased *p* level means an increased risk of the violation of environmental constraints. Meanwhile, it will lead to a decreased strictness for the constraints (and therefore, an expanded decision space, such as increased waste treatment/disposal capacity). That is, a higher system benefit (under a high *p* level) represents an alternative with higher waste generation and a higher waste treatment/disposal capacity, while a lower system benefit (under a lower *p* level) represents an alternative with lower waste generation and a lower environmental capacity. Usually, planning with lower system benefit can guarantee that waste management requirements and environmental regulations will be met. In comparison, with the planning aimed at higher system benefit, these requirements may not be met. Therefore, with the increased *p* level, the reliability of meeting the waste treatment/disposal capacity requirements and environmental requirements would decrease.

### 4.4. Fuzzy Relationship between the Economic Objective and the Constraints

The results also indicate that the optimized λ values were within the range of [0.37, 0.69]. The λ value represents the possibility of satisfying all the objectives and constraints under given system conditions. The solutions corresponded to conservative strategies when their λ values tended to be the lower bound. In comparison, the solutions became more optimistic when their λ values tended to be the upper bound. The relationship between the λ values and the system benefit is shown in Figure 7. It can be seen that the λ value and the system benefit exhibited a positive correlation. The IFTSP-LUAM achieved the maximum degree of satisfaction (λ value) for the system objective and constraints under uncertainty. Under λ = 0.37, the system benefit was [6.28, 7.02] × 10^12^ RMB, whereas under λ = 0.69, the system benefit was [9.64, 10.95] × 10^12^ RMB. The λ values indicated the trade-off between the system benefit and all the constraints (including environmental constraints). Lower λ values would guarantee that all the requirements were met, thus resulting in stricter constraints and a lower system benefit. In comparison, higher λ values would lead to more flexible constraints and a higher system benefit. For example, higher available electricity power, water and soil erosion corresponded to higher λ values and offer a higher system benefit.

## 5. Conclusions and Future Outlook

In this study, an interval two-stage stochastic fuzzy land-use allocation model (IFTSP-LUAM) was proposed for land-use planning and ecological environmental management of South Four Lake watershed. The proposed model was based on the interval two-stage stochastic fuzzy programming (IFTSP) model. The ITSFP model could effectively handle uncertainties expressed as discrete intervals, probabilities and fuzzy sets, and therefore, it could effectively support policy analysis, trade-off analysis and uncertain quantitative analysis. Furthermore, IFTSP-LUAM considered various economic, social, environmental and ecological factors in the land-use system, giving a series of land-use patterns, system benefits and ecological environmental protection strategies for sustainable development at a watershed level.

IFTSP-LUAM was applied to a land-use planning practice in South Four Lake watershed and obtained a series of results. These results could effectively support the government and decision-makers in formulating appropriate policies for land-use planning and ecological environmental management in South Four Lake watershed. Furthermore, this case study has proven the effectiveness, superiority and practicability of the IFTSP-LUAM. On the basis of fully considering the development conditions and objectives of the research area and the difficulty of data collection, the model can be applied to the corresponding parameters and conditions of other watersheds around the world.

The study of land-use allocation models will be endless because of the complexity of the land-use systems. Firstly, the IFTSP-LUAM has been successfully applied to a land-use allocation problem at a watershed scale. However, the application of this model to larger scales, such as national, large-area and global scales, needs to be studied. Secondly, the two-stage planning method could be improved as a multistage planning model for multisite land-use allocation. Thirdly, a complete land-use allocation model that considers more constraints and ecological factors needs to be researched. 

In the future, we will consider combining the quantitative allocation model IFTSP-LUAM with a spatial layout model, such as GIS, CA or other spatial optimization allocation models, and propose a coupled land-use optimization model that can simultaneously optimize the quantity and spatial allocation. Moreover, we have begun to combine the model of this paper with intelligent algorithm models, such as neural network, genetic algorithm, particle swarm optimization, ant colony algorithm and other intelligent algorithms, to develop the land use optimization model based on intelligent algorithms.

## Figures and Tables

**Figure 1 ijerph-18-13411-f001:**
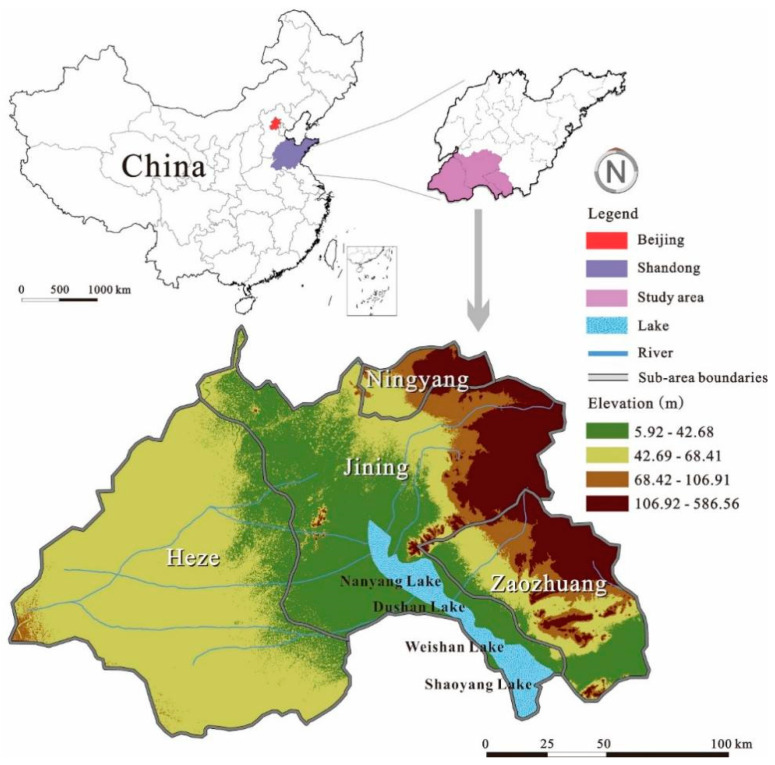
The study area.

**Figure 2 ijerph-18-13411-f002:**
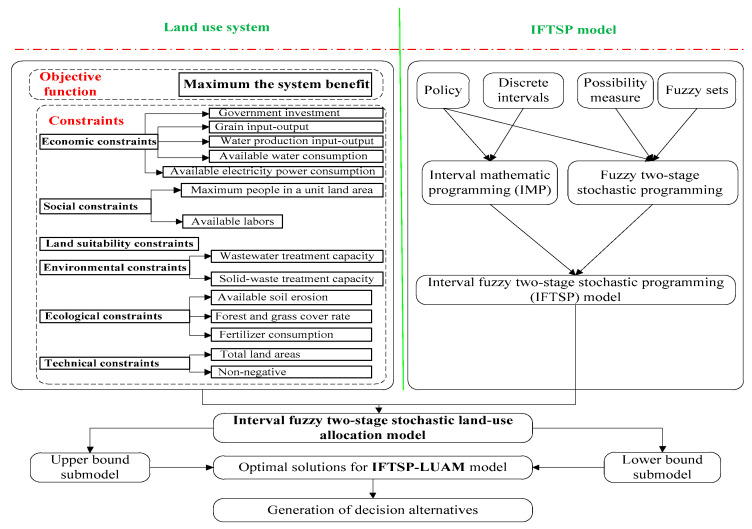
Framework of IFTSP-LUAM.

**Figure 3 ijerph-18-13411-f003:**
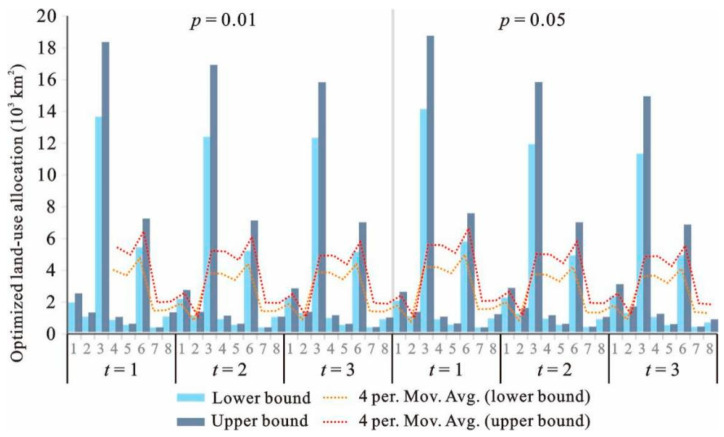
Optimized land-use allocation under *p* of 0.01 and 0.05 (*k* = 1).

**Figure 4 ijerph-18-13411-f004:**
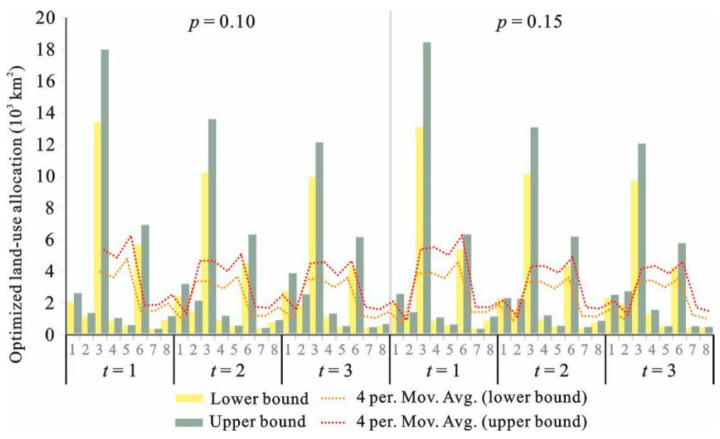
Optimized land-use allocation under *p* of 0.10 and 0.15 (*k* = 1).

**Figure 5 ijerph-18-13411-f005:**
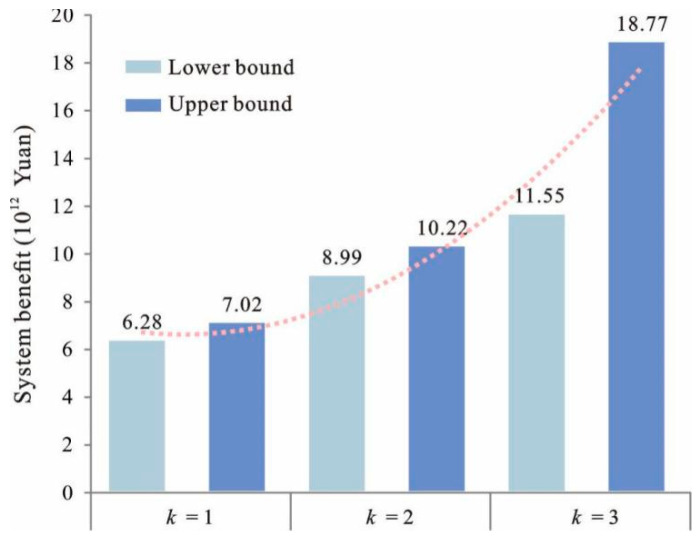
Relationship between land suitability level and system benefit.

**Figure 6 ijerph-18-13411-f006:**
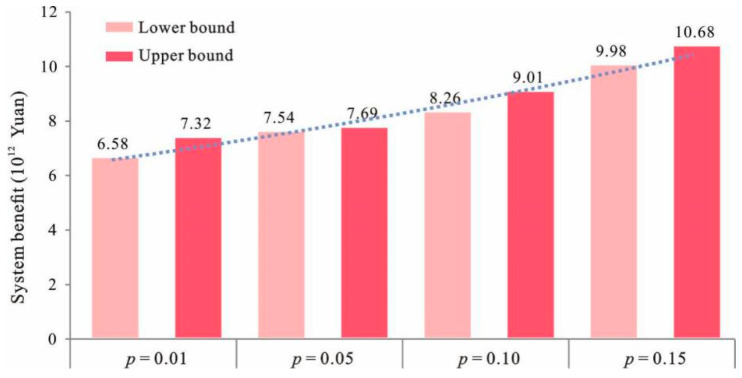
Relationship between *p* levels and system benefit.

**Figure 7 ijerph-18-13411-f007:**
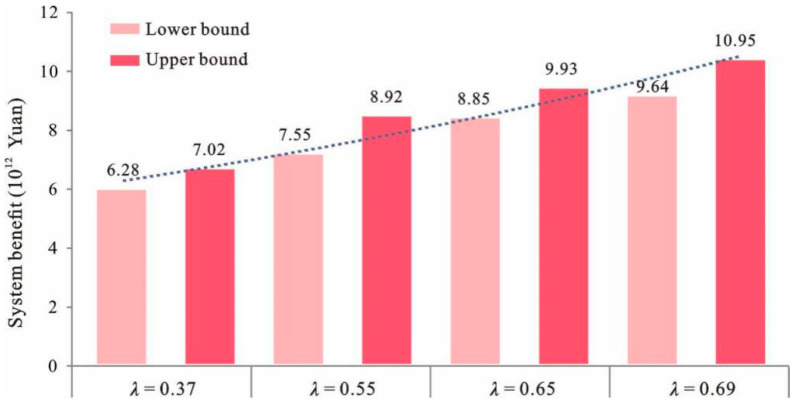
Relationship between λ and system benefit.

**Table 1 ijerph-18-13411-t001:** Description of the model.

Objective Function	Maximum Economic Benefit from the Land-Use System
Economic constraint	government investment should afford the system cost
	grain production should meet the demand [65]
	water production should meet the demand [66]
	water consumption of all land-uses should not exceed the available water supply [67]
	electricity power consumption of all land-uses should not exceed the available electricity power supply
Social constraint	maximum population should not exceed the land carrying capacity (LCC)
	planning labor should not exceed the available labor
Land-use suitability constraint	maximum land areas of each type of land use should accord with the results of the land suitability assessment [68]
Environmental constraint	wastewater should not exceed the wastewater treatment capacity
	solid waste should not exceed the solid waste treatment capacity and solid waste handling capabilities of the landfill
	air pollutants should not exceed the discharge limits [69]
Ecological constraint	planning agricultural land soil erosion should not exceed the available soil erosion area
	fertilizer consumption should not exceed the maximum fertilizer consumption [70]
Technical constraint	the sum of the allocated land area is the total land area of the study area
	the independent variable cannot be negative

**Table 2 ijerph-18-13411-t002:** Descriptions of the symbols.

Symbol	Descriptions	Symbol	Descriptions
NBL	The objective function, which represents the net benefit from land-use system of South Four Lake watershed (RMB)	* x *	The independent variable, which means land areas of each land use
RMB	Renminbi (RMB) is the legal currency of China	UB	The unit benefit of various types of land-uses *j* = 1–6 (RMB/km^2^)
±	Discrete interval values	≅	Fuzzy equal
USTC	The unit solid-waste-tackling cost of various land-uses *j* = 1–5 (RMB/km^2^)	UGTC	The unit waste-gas-tackling cost of various land-uses *j* =1–5 (RMB/km^2^)
UWTC	The unit wastewater tackling cost of various land-uses *j* = 1–5 (RMB/km^2^)	UWSC	The unit water-supply cost of various land-uses *j* = 1–5 (RMB/km^2^)
UMC	The unit maintenance cost of various land-uses *j* = 6–7 (RMB/km^2^)	UESC	The unit electric power-supply cost of various land-uses *j* = 1–5 (RMB/km^2^)
* k *	The land suitability condition, where *k* = 1 represents highly suitable, *k* = 2 represents moderately suitable, *k* = 3 represents lowly suitable	UDC	The unit developing costs of unused land (RMB/km^2^)
* t *	The planning period, where *t* = 1 for the time period of 2021–2025, *t* = 2 for the time period of 2026–2030, *t* = 3 for the time period of 2031–2035	* j *	The type of land-use, where *j* = 1 for commercial land, *j* = 2 for industrial land, *j* = 3 for agricultural land, *j* = 4 for transportation land, *j* = 5 for residential land, *j* = 6 for water land, *j* = 7 for landfill, and *j* = 8 for unused land

**Table 3 ijerph-18-13411-t003:** Optimized land-use patterns under different *p* levels at *t* = 1 and *k* = 1.

Land-Use Types	*p* Level			
	*p *= 0.01	*p *= 0.05	*p *= 0.10	*p *= 0.15
*j* = 1	[1608.8, 2176.6]	[1724.3, 2282.0]	[1743.8, 2348.3]	[1742.3, 2297.9]
*j* = 2	[721.7, 976.4]	[764.1, 1014.3]	[811.5, 1048.2]	[862.2, 1115.8]
*j* = 3	[13,303.0,17,998.1]	[13,772.5, 18,386.0]	[13,400.0, 18,074.4]	[13,088.3, 18,541.8]
*j* = 4	[515.2, 697.1]	[539.5, 718.2]	[555.2, 730.2]	[567.3, 766.4]
*j* = 5	[199.8, 270.4]	[216.3, 285.6]	[224.7, 269.2]	[229.4, 318.3]
*j* = 6	[5089.4, 6885.6]	[5448.6, 7212.8]	[5484.1, 6736.0]	[5186.1, 6139.9]
*j* = 7	[29.6, 40.1]	[27.9, 38.1]	[26.0, 35.4]	[23.9, 33.6]
*j* = 8	[730.6, 988.5]	[618.9, 872.8]	[599.0, 855.7]	[581.9, 812.9]

## Data Availability

Not applicable.

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
