# Peer review of "Sustainable Land-Use Allocation Model at a Watershed Level under Uncertainty"

_ijerph, 2021, doi:10.3390/ijerph182413411_

Round 1
Reviewer 1 Report
General comments:
The paper proposed an interval fuzzy two-stage stochastic land-use planning model (IFTSP-LUPM) for land-use allocation and ecological environmental analysis/management in South Four Lake watershed, Shandong, China. The first version of the paper was not clear to me. That version was improved; however it is not ready yet to be published. Some comments that I provide in the first revision were not considered, especially in the introduction section, that I still believe that it must be improved.
Some main comments are in the attached PDF.
I suggest revising the structure of the paper, especially in the section 2. The introduction section was revised, however not yet good. See the attached comments (in pdf file).
The tables must be cited in the text.
The equations were created/developed by the authors, or based on references? If the equations are based on literature, you should cite them. The authors do not clarify that.
References from 62 to 68 were not cited in the text.
More comments are in pdf attached.

Author Response
RESPONSES TO REVIEWER #1’ COMMENTS
We are very grateful to the reviewer for his/her insightful review. The comments and suggestions provided have contributed a great deal to improving the manuscript. According to these, we have made efforts in revising the manuscript, with the details explained as follows:
Note: For the reviewer’s convenience in re-reviewing, we have divided his/her comments into several parts with ordinal numbers, and responded to them on a point-by-point basis.RV: revised version.
According to your detailed comments and instructions in the PDF file, we have thoroughly revised the manuscript, as follows:
COMMENT 1 – Abstract: What type of models? Define the uncertainties?
RESPONSE: 1. We carefully modified the abstract according to your thorough comments in PDF, and all the words have been modified. Among them, there are two places that need special explanation:
What type of models?→RV: “At present, a large number of studies solved the problems of land use planning by constructing models,such as mathematical model, spatial analysis model and so on.”
Define the uncertainties?→RV: “the proposed IFTSP-LUAM takes into account the above three uncertainties”. The three uncertainties factors have been mentioned in the second sentence of the abstract.
COMMENT 2 – Keywords: Do not use words from title.
RESPONSE: We are thankful for the reviewer’s valuable suggestion.
RV: “Keywords: Sustainable land use;; Ecological environment protection; Interval fuzzy two-stage stochastic model; South Four Lake watershed; Eco-environmental constraints”
COMMENT 3 – I suggest revising the structure of the paper, especially in the section 2. The introduction section was revised, however not yet good. See the attached comments (in pdf file).
RESPONSE: We are thankful for the reviewer’s valuable suggestion. According to the detailed instructions in the PDF file, we have made the modifications.
COMMENT 4 –The tables must be cited in the text.
RESPONSE: All the tables have cited in the paper.
COMMENT 5 –This equation was created by the authors? If not give a reference.
RESPONSE: All the equations in the paper are constructed by ourselves.
Generally, we very much appreciate the reviewers’ insightful reviews. The provided comments/suggestions have contributed greatly to improving the manuscript.
Reviewer 2 Report
The paper proposed a land-use planning model (IFTSP-LUPM) for land-use allocation in Shandong, China's South Four Lake watershed.
There are still a lacks in the methods part, and there is a lot of information that should be a table part - that's why the text is still unclear. There is still the problem of equation presentation. The reader still does not know if the Authors presents the table or appendix in the text.
Thanks to the tables, the results part looks better than in the first round and is more precise.
There is still a lack of presentation on how authors can use the methods in other fields and territories. The sentence in the article is insufficient to correct this lack.
Author Response
RESPONSES TO REVIEWER #2’ COMMENTS
We are very grateful to the reviewer for his/her insightful review. The comments and suggestions provided have contributed a great deal to improving the manuscript. According to these, we have made efforts in revising the manuscript, with the details explained as follows:
Note: For the reviewer’s convenience in re-reviewing, we have divided his/her comments into several parts with ordinal numbers, and responded to them on a point-by-point basis.
COMMENT 1 – There are still a lacks in the methods part, and there is a lot of information that should be a table part - that's why the text is still unclear. There is still the problem of equation presentation. The reader still does not know if the Authors presents the table or appendix in the text.
RESPONSE: According to your valuable comments, we modified the methods part. Some information is shown in the table, and we clearly indicated the position of the table in the text.
COMMENT 2 – There is still a lack of presentation on how authors can use the methods in other fields and territories. The sentence in the article is insufficient to correct this lack.
RESPONSE:We put forward the prospect of this model in the future. It mainly includes: improving the accuracy and scientificalness of the model calculation, and combining this model with the land use spatial analysis. Please see it at L440-458.
Generally, we very much appreciate the reviewers’ insightful reviews. The provided comments/suggestions have contributed greatly to improving the manuscript.
Round 2
Reviewer 1 Report
General comments:
The paper proposed an interval fuzzy two-stage stochastic land-use planning model (IFTSP-LUPM) for land-use allocation and ecological environmental analysis/management in South Four Lake watershed, Shandong, China. The authors performed a revision of the paper. However still have some errors that should be corrected. Some comments that I provide in the previous revisions were not considered, especially in the introduction section.
Also the years of the study were changed. Why? Please clarify why changing your study data time period. Did you adopted your results to the new time period?
Some main comments are in the attached PDF.

Author Response
RESPONSES TO REVIEWER #1’ COMMENTS
We are very grateful to the reviewer for his/her insightful review. The comments and suggestions provided have contributed a great deal to improving the manuscript. According to these, we have made efforts in revising the manuscript, with the details explained as follows:
According to your detailed comments and instructions in the PDF file, we have thoroughly revised the manuscript, as follows:
COMMENT 1 – I give you some references that you should add to your study and the authors did not provide that references. I suggest to add.
RESPONSE: 1. We sincerely appreciate your comments. According to your suggestion, we have added the references about fuzzy programming and GIS. And we also added the references about random programming and interval programming.
Please it at References:[9-14] and [34-36].
COMMENT 2 – Legend must be more complete referring what is the figure.
RESPONSE: We are thankful for the reviewer’s valuable suggestion. We redesigned Figure 1. to make it more detailed.
Please see it at 258.
COMMENT 3 – We cannot have a section without text associated. Please give some information/description about the section content. Also cite equations 15 and 16.
RESPONSE: We are thankful for the reviewer’s valuable suggestion.
These are common problems in linear programming models. We neglected to explain this. Now, we have made some modifications and explanations.
“The sum of the allocated land area is the total land area of the study area”
“The independent variable cannot be negative”.
COMMENT 4 –Why did you changed these years? Do you changed your data in the study? Why?.
RESPONSE: This is the communication problem between the author who is responsible for calculation and the author who collects data when revising. It is actually 1995-2015.
Why take this period? There are two reasons:
Firstly, the parameter estimation of the model needs a lot of time series data, and we selected the data from 1995 to 2015.
Secondly, influenced by the development planning of different regions in the basin, the development of each region has changed greatly since 2016, and it is easy to generate more policy influence errors in data estimation. At the same time, affected by the COVID-19, the development since the end of 2019 is still in the recovery stage. Therefore, the data for 2016-2020 is not selected.
Generally, we very much appreciate the reviewers’ insightful reviews. The provided comments/suggestions have contributed greatly to improving the manuscript.

This manuscript is a resubmission of an earlier submission. The following is a list of the peer review reports and author responses from that submission.
Round 1
Reviewer 1 Report
The manuscript presents a sustainable land-use allocation model as an element of land-use policy. The case study is connected with the South Four Lake watershed area. The article is logical and presents step by step elements of the land use allocation model. The problem is that the authors must improve each equation's description by giving each measure in a new line.
The problem is that the authors write about Appendix 1, but it is missing, and authors must improve this part. The presentation of results must be improved because results, especially in chapter 4.1, are challenging to read and interpret by readers.
The authors could improve the last part of the manuscript by presenting how the model use in the research can be useful for different countries and different case studies. It means case studies not connected with watersheds but with regions or other research areas.
In my opinion, the article takes up an interesting research question and can be published after a few corrections presented in this review.
Reviewer 2 Report
General comments:
The paper proposed an interval fuzzy two-stage stochastic land-use planning model (IFTSP-LUPM) for land-use allocation and ecological environmental analysis/management in South Four Lake watershed, Shandong, China.
Some main comments are in the attached PDF.
In general, the English is good and understandable, the structure of the manuscript must be revised.
The abstract is confused. A lot of text but very few content. Reading the abstract I do not understand your work, because a lot of information is missing. Reformulate the abstract. Also you did not used GIS? Because you did not refer that in abstract.
In section 1.1., all the sentences gives me the same information, that all the authors described a multi-site land-use allocation model. But which one? What are the main differences between them? What are the advantages and disadvantages? You should be more precisely with the information you are giving.
In the sections 1.2, 1.3 and 1.4 the same comments as before.
The Introduction section requires a strong revision, considering my previous comment. You said that based on the literature, it can be inferred that a mathematical model is a good tool to support sustainable land-use planning. Why? I did not see the advantages and disadvantages of the models already existent. You should emphasize that, so the readers can understand your study and its importance.
You are presenting this model as it is new or not applied, however this reference https://www.sciencedirect.com/science/article/pii/S0377221704002929 presented the same methodology in 2005, even if applied in water resources, but you did not explain that. How can you explain that? Other study more recent https://pubmed.ncbi.nlm.nih.gov/31717718/ also present the same methodology applied to land use. So, your words are not correct. You should reformulate and clarify that.
All the sections with the equations must be in a table to better understanding. In the table, the acronyms must be defined below the table. Also, if the equations are based on literature, you should cite them.
More comments are in pdf attached.
I cannot accept the manuscript in this form. I will reject, but encourage the authors to reformulate and re-submit the manuscript.
